# RRescue: Ranking LLM Responses to Enhance Reasoning Over Context

## Abstract

Effectively using a given context is paramount for large language models (LLMs). A context window can include task specifications, retrieved documents, previous conversations, and even model self-reflections, functioning similarly to episodic memory. While efforts are being made to expand the context window, studies indicate that LLMs do not use their context optimally for response generation. In this paper, we present a novel approach to optimize LLMs using ranking metrics, which teaches LLMs to rank a collection of contextually-grounded candidate responses. Rather than a traditional full ordering, we advocate for a partial ordering. This is because achieving consensus on the perfect order for system responses can be challenging. Our partial ordering is more robust, less sensitive to noise, and can be acquired through human labelers, heuristic functions, or model distillation. We test our system's improved contextual understanding using the latest benchmarks, including a new multi-document question answering dataset. We conduct ablation studies to understand crucial factors, such as how to gather candidate responses, determine their most suitable order, and balance supervised fine-tuning with ranking metrics. Our approach, named RRescue, suggests a promising avenue for enhancing LLMs' contextual understanding via response ranking.

## 1 Introduction

A significant advantage of large language models (LLMs) is their ability to provide explanations, or rationales, for their predictions (Ziegler et al., 2020; Bai et al., 2022a; Chiang et al., 2023; Touvron et al., 2023). As these models increasingly assist in decision-making processes across domains, examining the quality of their rationales becomes crucial. For example, LLMs can recommend laboratory tests to physicians based on patient symptoms (Peng et al., 2023), or help financial analysts evaluate risks in their investment portfolios (Romanko et al., 2023), providing rationales for each. However, the quality of the rationales they provide can vary, and flawed rationales can lead to poor decisions and misinformation. Thus, it is essential to enhance the LLMs' ability to produce high-quality rationales, improving their overall accuracy in tasks.

The quality of rationales produced by LLMs is linked to their ability to reason over context. Their context window can include task specifications, retrieved documents, historical conversations, and more. LLMs need to reason over *relevant parts of the context* to generate valid rationales supporting their predictions. Relying on model's self-reflection mechanisms, such as chain- or tree-of-thoughts, can often be insufficient for developing sound rationales (Yao et al., 2022; Wei et al., 2023; Yao et al., 2023; Shinn et al., 2023). In domain-specific tasks, such as assigning risk levels to investment portfolios, this is particularly true, as humans may annotate tens of thousands of examples with expert rationales. It is thus vital to devise new methods to leverage limited expert annotations to enhance the models' contextual understanding.

As context size grows, LLMs can struggle to identify pertinent parts of their context, especially when dealing with multiple documents as their context. This challenge is highlighted by Liu et al. (2023), which suggests that LLMs' reasoning abilities notably degrade when relevant information is not positioned at the beginning or end of such context. It often leads to poorer results than if no context was used. Consequently, it is beneficial for LLMs to recognize whether their rationales are accurately grounded on the relevant context sections. This ability enables the model to better incorporate relevant contextual information during rationale generation.

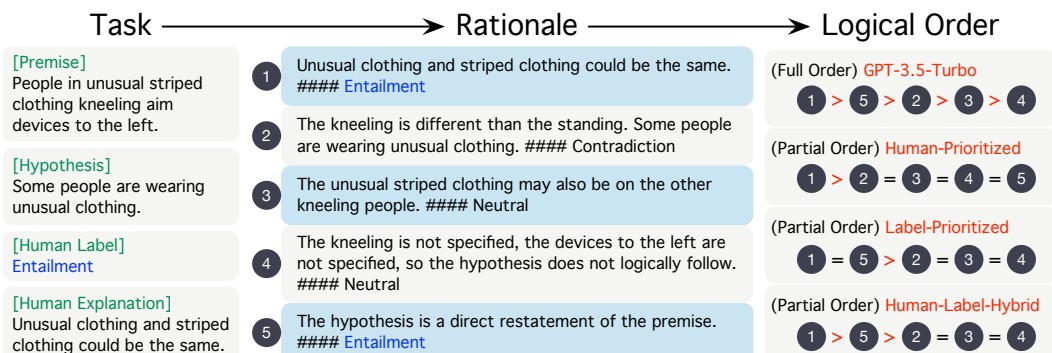

Figure 1: Our proposed framework. We aim to improve task accuracy by refining LLMs' reasoning abilities. Our method fine-tunes LLMs to effectively rank a set of candidate responses, which we acquire from diverse sources. Multiple partial orderings of responses are obtained through limited human annotations and heuristics. These partial orderings are not only more robust but also less sensitive to noise compared to traditional full orderings.

Our goal in this paper is to improve LLMs' task accuracy by refining their reasoning abilities. We focus on LLM responses that contain *a short answer or label*, accompanied by *a more detailed rationale*. It is crucial that LLM responses not only yield correct labels but also provide sound rationales supporting their predictions. We explore methods to fine-tune a recently released LLM to effectively rank a set of candidate responses, which we acquire from diverse sources. Ranking responses allows the LLM to differentiate between sound and flawed rationales, and recognize rationales that are grounded on the relevant context sections, and thus improve their reasoning skills.

Our approach is distinct from learning from human feedback, such as RLHF. Those methods require human preference judgments on millions of LLM response pairs, a task too resource-intensive for most research institutions. In contrast, we employ a more feasible approach, utilizing a partial ordering of LLM responses that can be acquired through limited human annotations, heuristic functions, or even model distillation. Our approach seeks to make the LLM more efficient for domain-specific tasks, emphasizing both accuracy and sound reasoning.

We test our system's improved reasoning abilities using benchmark datasets, including a new multi-document question answering dataset (Liu et al., 2023). We conduct a series of ablation studies to understand crucial factors, including how to gather diverse candidate responses, determine their most suitable order, and balance supervised fine-tuning with ranking metrics. We discuss challenges faced during our experiments and shed light on potential future directions. Our approach, named RRESCUE, presents a promising avenue for enhancing LLMs' contextual understanding via response ranking.[1]

## 2 RELATED WORK

**Learning from Human Feedback**    Aligning LLM responses with human values through learning from feedback ensures the models' outputs are helpful, safe, and adhere to societal norms (Bai et al., 2022b). This research involves humans performing pairwise or k-wise comparisons on model outputs, which are used to train a reward model (Ziegler et al., 2020; Bai et al., 2022a; Ouyang et al., 2022; Ramamurthy et al., 2023; Zhu et al., 2023). The reward model, combined with proximal policy optimization, rejection sampling, and other optimizers, refines LLMs' responses for general prompts. Unlike this approach, our study focuses on task-specific accuracy, such as predicting investment portfolio risk levels. We aim to guide LLMs to make accurate predictions and provide sound rationales using the limited expert annotations available for those tasks.

**Self-reflection** enables LLMs to enhance their reasoning by learning through trial and error and self-improvement. For example, chain-of-thought (Wei et al., 2023) allows LLMs to break down complex tasks step by step into more manageable parts. Tree of thoughts (Yao et al., 2023) employs task decomposition via a tree structure, guiding LLMs through various steps and consider multiple thoughts within each step. Reflexion (Shinn et al., 2023) combines dynamic memory and self-reflection to refine reasoning skills. It also adds reflections into the agent's working memory as context for querying LLM (Dubois et al., 2023). However, pinpointing specific reasoning errors

---

[1]We will make our source code publicly available to the research community.

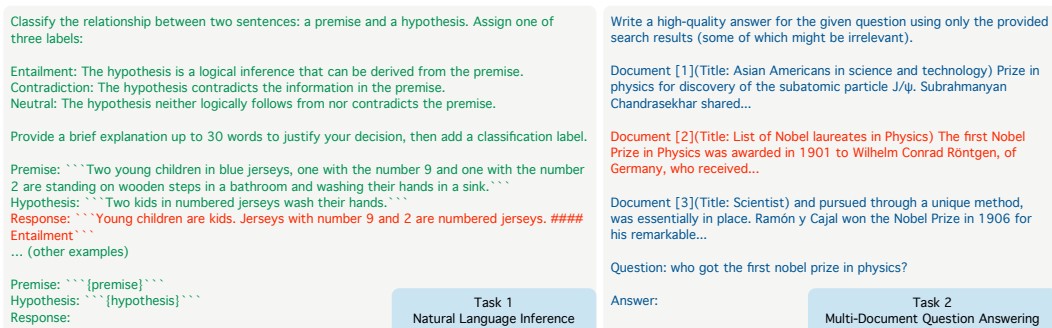

Figure 2: (LEFT) For the natural language inference task, candidate responses are solicited from the Llama-2 and ChatGPT model, each consisting of a short label and a detailed rationale, separated by '####'. (RIGHT) For the multi-document question answering task, diverse responses are obtained by providing one supporting document at a time. Responses generated from reference documents (colored in red) receive priority over other candidates.

remains a practical challenge. The distinction between sound and flawed rationales can often be subtle and unclear during self-reflection.

**Ranking Metrics**     A ranking objective allows the model to prioritize reference output over alternative candidates, improving its performance in tasks like abstractive summarization and question answering. For example, the BRIO training paradigm (Liu et al., 2022) fine-tunes BART and T5 models to generate reference summaries while using a ranking mechanism to score candidate summaries. This approach, similar to contrastive learning, could be especially beneficial in retrieval augmented generation (Hopkins & May, 2011; Lewis et al., 2021; Nakano et al., 2022). We conjecture that rationales grounded on falsely retrieved documents should be discounted compared to those grounded in reference documents. Our method, inspired by BRIO, extends it to label prediction and rationale generation, improving the model's capability to produce contextually accurate rationales.

**Model Distillation** involves training a compact student model to replicate the behavior of a more complex teacher model. It allows the distillation of large models into more lightweight versions, facilitating deployment in resource-constrained scenarios (Magister et al., 2023; Hsieh et al., 2023). Given the significant GPU memory and specialized infrastructure required to serve large LLMs with hundreds of billions of parameters, model distillation is gaining attention for its potential to provide smaller, task-specific models. Our study utilizes responses from ChatGPT and other state-of-the-art LLMs for distillation. Unlike other studies focused on distillation, our approach doesn't just distill knowledge from a single LLM; it leverages on rationales from multiple LLMs to train a compact model that produces enhanced task performance.

## 3   OUR APPROACH

Let $x \sim \mathcal{D}$ represent the prompt or context given to the model, and $y$ denote the model's response to prompt $x$. The response $y$ comprises two parts: a brief justification and a predicted label, e.g., "*Unusual clothing and striped clothing could be the same. #### Entailment*" in the natural language inference task. Supervised fine-tuning (SFT; Eq. (1)) is a primary method to improve task accuracy by refining the model to produce human-labeled responses $y^*$. However, since the model has only been exposed to high-quality human responses, its noise robustness remains untested. Prior studies (Ziegler et al., 2020; Touvron et al., 2023) suggest that model performance can plateau quickly, potentially leading to overfitting.

$$\mathcal{L}_{\text{SFT}}(\theta) = -\log \pi_\theta(y^*|x) \tag{1}$$

$$\mathcal{L}_{\text{Rank}}(\theta) = -\mathbb{E}_{(x,y_0,y_1,b)\sim\mathcal{S}}\big[\max\{0, \log \pi_\theta(y_b|x) - \log \pi_\theta(y_{1-b}|x)\}\big] \tag{2}$$

$$\mathcal{L}_{\text{RRESCUE}}(\theta) = \mathcal{L}_{\text{SFT}}(\theta) + \alpha\mathcal{L}_{\text{Rank}}(\theta) \tag{3}$$

We aim to learn from *preference* data, rather than *reference* data. We guide the model to prioritize valid responses over flawed ones, and those that are contextually accurate over falsely grounded ones, using a ranking metric as illustrated in Eq. (2). Here, $(x, y_0, y_1, b) \sim \mathcal{S}$ contains a prompt $x$ and two candidate responses, with $y_b$ to be scored higher than $y_{1-b}$. $\mathcal{S}$ represents a diverse set of candidate responses obtained from various sources. For example, responses could be acquired from open-

You are a helpful assistant. When given an input followed by its respective label, please rank the following five candidates based on their semantic similarity to the input. The label can be either Entailment, Neutral, or Contradiction. Candidate with the same labels as the input should be ranked higher than those with different labels. Only show the ranks; no justification is needed.

**Full Order Provided by GPT**

Input: Unusual clothing and striped clothing could be the same. #### Entailment

Candidate 1. The people in the unusual striped clothing are wearing clothing. #### Neutral
Candidate 2. The people are wearing unusual clothing and aiming devices to the left. #### Entailment
Candidate 3. The clothing is not mentioned in the premise. #### Neutral
Candidate 4. The people in the first sentence are wearing unusual striped clothing. The people in the second sentence are wearing unusual clothing. #### Neutral
Candidate 5. People are wearing unusual clothing while kneeling aim devices to the left. #### Entailment

Figure 3: Our approach "(FO) GPT-3.5-Turbo" leverages the GPT-3.5-Turbo-0613 model to organize candidate responses into a strict order. We instruct it to prioritize candidates with the same labels as the human response.

source LLMs like Llama-2 or close-source LLMs like GPT-3.5, GPT-4 or Claude. Human-annotated responses can also be included in the collection whenever they are available.

We initiate $\pi_\theta(y|x)$ from a base model $\rho(y|x)$ and subsequently fine-tune it for a specific task with candidate responses. Particularly, $\pi_\theta(y|x)$ is used to loosely represent length-normalized probability $\pi_\theta(y|x) = \frac{1}{|y|^\lambda} \sum_{t=1}^{|y|} \log \pi_\theta(y_t|x, y_{<t})$, where $\lambda > 0$ is the scaling factor for length normalization. Our approach, RRESCUE, uses a hyperparameter $\alpha$ to balance the impact of supervised fine-tuning and the ranking metric (Eq. (3)).

**Organizing LLM Responses**  Candidate responses $\{y_i\}_i$ for a given prompt $x$, can be organized into a strict order. For example, InstructGPT utilizes a team of trained human labelers to rank sets of model outputs from best to worst to train a reward model (Ouyang et al., 2022). However, this method is too costly for most research institutions. We propose two alternative cost-effective approaches to establish a Full Ordering (FO) of responses.

Our first approach, **(FO) Similarity**, embeds each candidate response into a vector, which are then ranked based on their Cosine similarity to the vector representing the human response. Our second approach **(FO) GPT-3.5-Turbo** leverages the GPT-3.5-Turbo-0613 model to rank candidate responses. We instruct it to prioritize candidates with the same labels as the human response, but allowing it to decide whether this criterion is met. An illustration of our prompt is given in Figure 3.

Conversely, Partial Orderings (PO) of responses offer enhanced flexibility and noise robustness. For example, in developing Llama-2, Touvron et al. (2023) noted that even human labelers struggle to decide between two similar model responses, with annotations for such responses often hinging on subjective judgement and nuanced details. By utilizing a partial order, we only incorporate the most clear-cut pairs of model outputs in the ranking metric, thereby improving the quality of response pairs used in model fine-tuning.

Our method, **(PO) Human-Prioritized**, posits that human responses should take priority over model responses, as they offer valid rationales and accurate labels. Similarly, **(PO) Label-Prioritized** places responses with correct labels above those with incorrect labels, irrespective of whether they are human or model-generated. This is because rationales resulting in correct labels are more valuable than those leading to incorrect labels. The latter may contain flawed reasoning that misguides their predictions. We note that only human labels are needed to establish this partial order.

Lastly, **(PO) Human-Label Hybrid** employs a fine-grained grouping. It places human responses above model responses with correct labels, which are then prioritized over responses with incorrect labels. This hierarchy is designed to motivate the LLM to generate rationales comparable to humans' or, at a minimum, to produce rationales that lead to accurate labels.

$$\mathcal{L}_{\text{Reward}}(r) = -\mathbb{E}_{(x,\{y_i\}_i,b)\sim\mathcal{S}}\left[\log \frac{e^{r(x,y_b)}}{\sum_i e^{r(x,y_i)}}\right] \tag{4}$$

**Comparing Our Ranking Metrics and Reward Models**  A reward model $r(x, y_i)$ assigns scores to a given prompt $x$ and its corresponding response $y_i$. Ziegler et al. from OpenAI pioneered the reward model, illustrated in Eq. (4), which allocates the *full probability mass* to the response $y_b$ chosen by human labelers. For this model to function, humans need to provide accurate pairwise preference judgments. Nonetheless, achieving a consensus among human labelers regarding the perfect order of

| Training Data | RRESCUE | | | | w/ Flip | |
|---|---|---|---|---|---|---|
| | 0.4% | 0.9% | 1.8% | 3.6% | 0.4% | 0.9% |
| (SFT) Supervised Finetuning | 77.45 | 85.56 | 87.33 | 87.94 | - | |
| (PO) Human-Prioritized | 80.70 | 87.11 | 87.06 | 86.26 | 86.80 | 85.81 |
| (PO) Label-Prioritized | 81.97 | 87.27 | **88.16** | **87.97** | 87.12 | **87.88** |
| (PO) Human-Label Hybrid | **82.86** | **87.47** | 87.33 | 87.73 | **87.74** | 87.81 |
| (FO) Similarity | 81.01 | 86.69 | 86.53 | 86.38 | 86.19 | 86.43 |
| (FO) GPT-3.5-Turbo | 82.20 | 86.62 | 85.02 | 86.71 | 85.29 | 85.30 |

Table 1: Task accuracy of RRESCUE on natural language inference, reported on the e-SNLI test set. Our model is fine-tuned with an extremely limited amount of training data, ranging from 0.4% to 3.6% of the original training set, i.e., between 1k to 20k examples. We observe that both partial orderings (PO) approaches—label-prioritized and human-label-hybrid—demonstrate competitive performance, comparable to the state-of-the-art results. E.g., Hsieh et al. (2023) reported a best accuracy of 89.51% for this task using a 540B LLM for step-by-step distilling. We find that utilizing flipped responses does not yield improved performance. Further, supervised fine-tuning (SFT) tends to yield better accuracy when more training data is available. This could possibly be associated with dataset artifacts, where the labels can be derived by looking only at the hypothesis (Gururangan et al., 2018). We point out that Llama-2-7B only attains an accuracy of 33.31% in a zero-shot setting, without any fine-tuning.

LLM responses can be a daunting task. The labelers often struggle to provide consistent, fine-grained labels (Touvron et al., 2023). As a result, allocating the entire probability mass, i.e., $\log \mathcal{P}_\theta(y_{b'}|x)$ to an incorrectly labeled response $y_{b'}$ can mislead the model and hinder the effective training of the reward model.

In contrast, our proposed ranking metrics offer greater flexibility and robustness to inconsistencies in human feedback. The model not only prioritizes $y_b$ over other potential responses with the equation $\max\{0, \log \mathcal{P}_\theta(y_b|x) - \log \mathcal{P}_\theta(y_{1-b}|x)\}$, but further allows minor deviations. For example, the model can still assign a high probability to a less-favored response $\log \mathcal{P}_\theta(y_{1-b}|x)$, provided its probability difference from the top response $\log \mathcal{P}_\theta(y_b|x) - \log \mathcal{P}_\theta(y_{1-b}|x)$ remains minimal. We also advocate for a partial ordering of LLM responses, partitioning them into groups. This group ordering provides a hierarchical perspective, enabling the model to understand the relative importance of each group in a broader context.

## 4 COLLECTING LLM RESPONSES

We obtain diverse model responses for the natural language inference task from both open-source $y \sim \mathcal{P}_{\text{Llama-2}}(y|x)$ and closed-source $y \sim \mathcal{P}_{\text{GPT}}(y|x)$. Specifically, we sample three responses from LLama-2-7B with a temperature value of 0.8 to encourage diverse outputs, one from GPT-3.5-Turbo-0613, along with the human response, totaling five candidate responses per prompt. Each response includes a concise rationale showing the reasoning, followed by a predicted label. The LLMs are given a task prompt with three in-context examples, as illustrated in Figure 2.

**Flipping Responses** We propose a novel method to elicit diverse and natural responses from LLMs. For example, we can flip a response, such as "*The to-go packages may not be from lunch. #### Neutral*," to "*The to-go packages are likely from lunch. #### Entailment*" to ensure diverse reasoning. To do this, we invert the meaning of a model-generated rationale using GPT, concatenate it with the original context, and then prompt GPT to predict the corresponding label for the altered rationale. We employ GPT-4 to invert rationales, given its solid performance. The prompt is "*Rewrite the sentence to convey the opposite meaning: Sentence.*" For predicting the new label, we use GPT-3.5-Turbo due to its cost-effectiveness. This method allows us to generate a sizable collection of candidate responses featuring varied reasoning.

We also explore multi-document question answering (Liu et al., 2023). Here, the model receives a question along with multiple Wikipedia documents; one document has the answer, and the rest are distractors. To generate candidate responses for this, we prompt Llama-2 (see Figure 2) to ground its answer on a single document in the context. If the reference answer phrase is in the model's response, we label it as correct. For example, candidate answers such as "*The first Nobel Prize in Physics was awarded in 1901 to Wilhelm Conrad Röntgen for his discovery of X-rays. #### Correct*" and "*Ernest Lawrence was the recipient of the first Nobel Prize in Physics. #### Incorrect*" are generated for this task. We generate five responses from Llama-2, grounding the responses on one reference document and four random documents. Human responses are unavailable for this task.

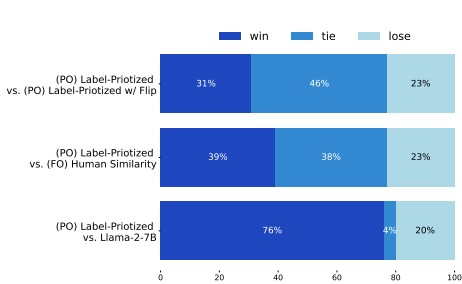
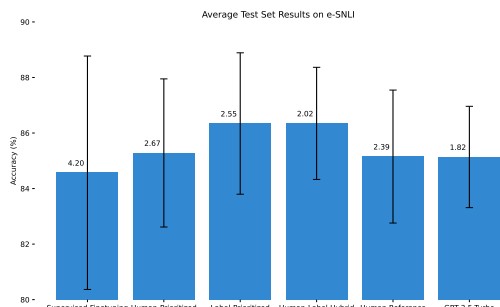

Figure 4: (LEFT) human evaluation for RRESCUE. We evaluate the partial ordering (PO) Label-Prioritized approach in comparison to itself using flipped responses (top bar), the full ordering (FO) Similarity approach (middle bar), and the Llama-2-7B base model (bottom bar). We observe that RRESCUE significantly surpasses the base model and also holds an edge over the full ordering approach. The difference in using flipped responses is not prominent. (RIGHT) the task accuracy of all approaches on the e-SNLI test set, averaged over all training data sizes, with the standard deviation included. SFT shows the widest performance range across all sizes. The partial ordering (PO) approach with a Human-Label Hybrid attains the highest average task accuracy.

## 5 EXPERIMENTS

We evaluate our proposed approach using two benchmark datasets: e-SNLI (Camburu et al., 2018) and the latest multi-document question answering dataset (Liu et al., 2023). The Stanford Natural Language Inference dataset (SNLI) presents this as a classification task: given a pair of sentences, the premise and hypothesis, their relationship is categorized as entailment, contradiction, or neutral. The e-SNLI dataset augments SNLI by including human-annotated explanations, forming tuples of (premise, hypothesis, label, explanation). These explanations address the question "*Why is a pair of sentences categorized as entailment, neutrality, or contradiction?*" Given its popularity, we can directly compare our findings with state-of-the-art results.

### 5.1 EXPERIMENTAL SETTINGS

**Base Language Model**    Meta's Llama model series leveraging auto-regressive transformers is a leader in open-source LLMs. Llama-2 is the latest development in this series, outperforming other open-source models such as Falcon (Penedo et al., 2023), BLOOM (Scao et al., 2022), Vicuna (Chiang et al., 2023), and MPT (MosaicML, 2023). We opt for the LLAMA-2 version over LLAMA-2-CHAT in this study due to our focus on non-dialogue tasks. Moreover, we select the more compact Llama-2-7B variant with the goal of enhancing task performance without the need for extensive GPU memory or specialized infrastructure to serve the model.

**Hyperparameter Settings**    We employ the AdamW optimizer (Loshchilov & Hutter, 2017) with a learning rate of $2e^{-5}$, combined with a cosine learning rate scheduler with a warmup rate of 0.03. To expedite LLM training, we apply fully sharded data parallelism and use BF16 for mixed precision training. BF16 is generally faster, consumes less memory, and is preferable for large neural models. Our experiments are conducted using 4 A100 GPUs, and model fine-tuning is constrained to a single epoch for both supervised fine-tuning and ranking metrics. This is to mitigate the risk of multi-epoch degradation (Xue et al., 2023) and potential overfitting from repeated exposure to the training data.

**Batch Size**    We use a batch size of $B$=64, following the fine-tuning settings of LLama-2. The batch size is the product of three factors, $B = g \times b \times D$, where $g$ is the gradient accumulation steps, $b$ is the batch size per GPU device, and $D$ is the number of devices used. Due to hardware memory limitations, we set the per-device batch size to one to accommodate a large number of responses during the ranking. We accumulate gradients for 16 steps before updating the model. In the right figure, we show preliminary experiments with varying batch sizes where we sample 10k instances repeatedly for supervised fine-tuning. Results suggest that increased batch size stabilizes training, reduces variation, and slightly improves task accuracy.

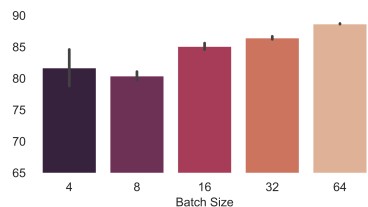

Figure 5: Varying batch sizes. We sample 10k instances repeatedly for supervised fine-tuning.

| Golden Position | 5 documents | | | | 10 documents | | | |
|---|---|---|---|---|---|---|---|---|
| | 0 | 2 | 4 | Average | 0 | 4 | 9 | Average |
| Llama-2-7B Base | **45.64** | 34.19 | 43.05 | 40.96 | **46.41** | 27.17 | 42.95 | 38.84 |
| (PO) Label-Prioritized | 44.88 | **42.44** | **53.43** | **46.92** | 35.72 | **33.43** | **55.11** | **41.42** |

Table 2: Task accuracy on multi-document question answering. The task involves answering a given question using a set of retrieved documents (Liu et al., 2023). Two settings are under consideration: the model receives 5 documents, or 10 documents, returned by the retriever. The reference document is placed at the beginning (position 0), middle (position 2 or 4), or the end (position 4 or 9) among all documents. We report answer accuracy on a test set containing 665 examples, following the guidelines provided by Liu et al. (2023). RRESCUE is fine-tuned to recognize rationales grounded on the reference documents. It achieves substantial performance gain when the reference document is positioned in the middle or end of the document set.

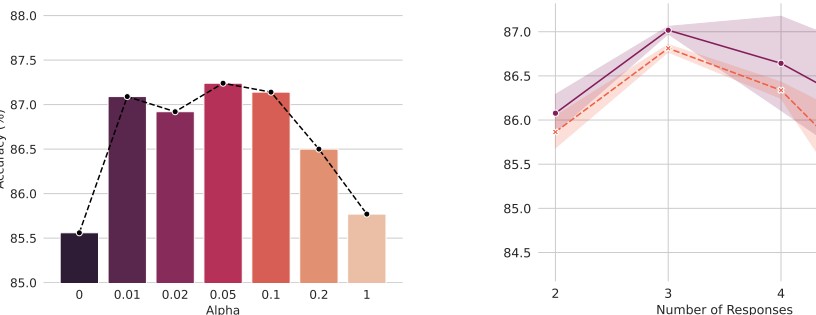

Figure 6: (LEFT) The influence of different $\alpha$ on task accuracy. We find that optimal performance is achieved with an $\alpha$ value between 0.01 to 0.1. (RIGHT) We conduct experiments with a varying number of candidate responses per prompt. Results indicate that performance improvement can be achieved even with 3-4 candidate responses. Beyond that, RRESCUE sees no further gains from increasing the number of responses. This saturation in performance may be attributed to the noise in ranking. Moreover, it highlights the challenges associated with ranking a diverse set of responses differing in length and style of rationales.

## 5.2 RESULTS ON TEXTUAL ENTAILMENT

**Data Sizes** The e-SNLI dataset contains 550k instances for the training set, 9842 and 9824 instances for the validation and the test set, respectively. Given the computational resource constraints associated with fine-tuning a Large Language Model (LLM) on the entire e-SNLI training set, which comprises 549,367 examples, we choose to utilize a fraction of it. As illustrated in Table 1, we evaluate the performance on varying proportions and report the corresponding outcomes.

**SFT** We conduct supervised fine-tuning and report the results of this approach on the e-SNLI dataset, leveraging the human-annotated responses available within the dataset. In the original dataset, both the ground truth label and the human-annotated explanation are provided, which we concatenate for our study. Our observation suggests a rapid accuracy increase with a rise in training data size, even when limited to a small sample such as 5k examples. However, we notice a performance plateau when we further scale the sampled training dataset.

**RRescue on e-SNLI** We validate the accuracy of the fine-tuned Llama-2-7B model using our method and compare its performance with the base model as well as a previous SOTA method. This comparison is conducted under various training data sizes, as illustrated in Table 1. Our results show that our method performs comparably to state-of-the-art approaches, a noteworthy achievement given its use of only a small portion of the full training set. When training data is limited, our method variants yield benefits over the Supervised Fine-Tuning (SFT) approach. However, as data size increases, performance is constrained by ranking noise.

Our findings suggest that for both the Partial Order (PO) and Full Order (FO) approaches, the benefits from expanding the training data eventually reach a plateau, with this saturation point occurring later for the PO method. This indicates that the PO approach is more robust to noise. In terms of overall performance, the PO method outperforms the FO method. Interestingly, we note the Llama-2-7B (Touvron et al., 2023) base model exhibits significantly low prediction accuracy. For a more in-depth discussion of it, refer to 5.4.

**Human Evaluation** Figure 4 presents a comparison of response quality under various settings, utilizing 100 pairs for an intuitive model assessment akin to Touvron et al. (2023). The assessment

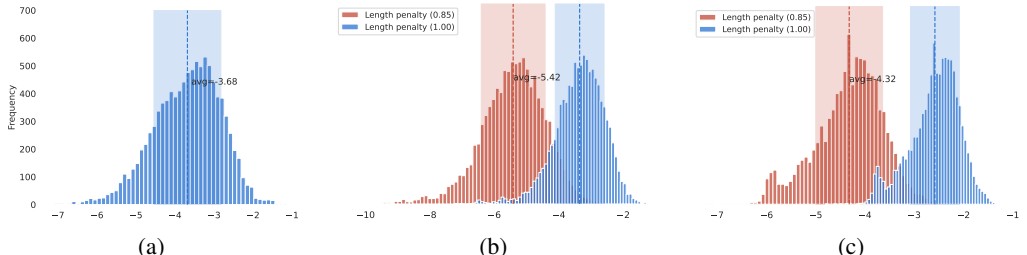

Figure 7: 7(a) shows the log probabilities of human responses, while 7(b) and 7(c) present those from Llama-2-7B and GPT-3.5-turbo-0613, respectively. We assign a length scaling factor, $\lambda$, of 0.85 to all model responses, maintaining a $\lambda$ of 1.0 for human responses. This approach effectively shifts the log probability score distributions of model responses (colored in red) closer to those of human ones, thereby minimizing margin violations.

prioritizes the accuracy of the predicted label and its corresponding explanation, and considers the overall text quality. A human annotator judges the responses, choosing from win, tie, or lose.

We evaluate the Partial Order (PO) model against three other models: PO with response flipping, Full Order (FO), and the base model. The model with the highest accuracy for each method is selected, as detailed in Table 1. The PO model surpasses its counterparts in human evaluation, with the performance difference becoming more pronounced when compared with the FO and the base model.

**Response Flipping**  By employing the method detailed in Section 4, we enhance our data diversity by flipping responses on e-SNLI, derived from a variety of sources. This procedure produces a dataset of flipped responses, providing richer information for comparing different variants of our method, as presented in Table 1. The results demonstrate that PO (Partial Ordering) exhibits higher accuracy compared to FO (Full Ordering). This finding further validates the notion that partial ordering is more robust and yields higher scores.

### 5.3 Results on Multi-Document Question Answering

The task involves answering a given question using a set of retrieved documents (Liu et al., 2023). During training, we generate five candidate responses per instance from Llama-2, grounding the responses on one reference document and four random documents. We sample 1k instances from the training data for fine-tuning. Since human responses are unavailable for this task, we fine-tune the model exclusively using our ranking metric, as shown in Eq. (2), without resorting to supervised fine-tuning (Eq. (1)). At test time, we consider two settings: the model receives either 5 or 10 documents returned by the retriever. The reference document is placed either at the beginning (position 0), in the middle (position 2 or 4), or at the end (position 4 or 9) of all documents.

In Table 2, we report answer accuracy on a test set containing 665 instances, following the guidelines provided by Liu et al. (2023). RRESCUE is fine-tuned in this task to recognize rationales grounded on the reference documents. It achieves substantial performance gain when the reference document is positioned in the middle or end of the document set. After closely examining the model's responses, we find the model often answer questions by copying content, particularly from the last document in the context. This tends to improve answer accuracy, especially when the reference document is positioned at the end of the context.

### 5.4 Discussions

**Balancing Coefficient and Number of Candidate Responses**  Our approach, RRESCUE, uses a hyperparameter $\alpha$ to balance the impact of supervised fine-tuning and the ranking metric (Eq. (3)). Figure 6 shows the influence of different $\alpha$ on task accuracy. We find that optimal performance is achieved with an $\alpha$ value between 0.01 to 0.1. The results indicate that, while supervised fine-tuning is pivotal for RRESCUE, integrating the ranking metric enhances the method's robustness to noise.

We conduct experiments with a varying number of candidate responses per prompt, and the results are shown in Figure 6. In our experiments, we are able to rank up to five candidate responses using four Nvidia A100 GPUs. As the number of candidates increases, so does the demand for additional GPU memory and compute resources. Our experiments indicate that performance improvement can

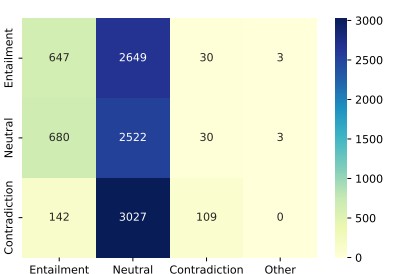 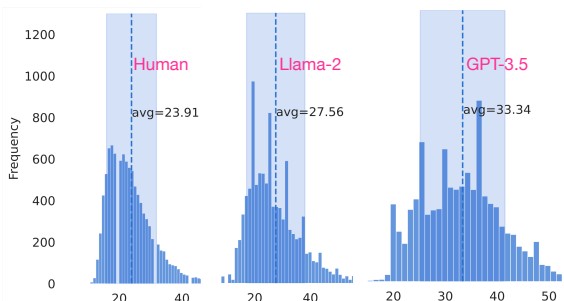

Figure 8: (LEFT) The confusion matrix for the Llama-2-7B base model, where the x-axis represents the labels predicted by Llama-2-7B, and the y-axis represents human labels. The results show Llama-2-7B's tendency to predict neutral labels, as indicated by the dark bar in the middle. (RIGHT) Candidate responses differ in length. We show the distribution of responses from human annotators, Llama-2-7B, and GPT-3.5-turbo-0613 models. Human responses are the shortest, while GPT-3.5's are notably longer, containing on average 10 more tokens per response compared to human responses.

be achieved even with 3-4 candidate responses. Beyond that, RRESCUE sees no further gains from increasing the number of responses. This saturation in performance may be attributed to the noise in ranking. Moreover, it highlights the challenges associated with ranking a diverse set of responses differing in length and style of rationales.

**Central tendency bias**    We notice that LLMs such as Llama-2-7B and GPT-3.5 exhibit a central tendency bias (Goldfarb-Tarrant et al., 2020) in natural language inference. These models often predict *Neutral* labels, leaning towards the "center" of possible labels. Figure 8 presents the confusion matrix, with the x-axis representing predicted labels by Llama-2-7B and the y-axis showing human labels. The results show Llama-2-7B's tendency to predict neutral labels (indicated by the dark bar in the middle) and its avoidance of extreme labels like *Entailment* or *Contradiction*.

A plausible reason could be Llama-2-7B's inadequate world knowledge impacting its task accuracy. Moreover, this tendency might originate from the models being trained on human annotations for instruction-following. They frequently give hedging responses to fulfill helpfulness and safety requirements, leading to outputs that are more neutral and less assertive.

**Scoring Candidate Responses**    We identify two characteristics in human responses that distinguish them from model responses. Firstly, they are more concise and to the point. As indicated in Figure 8 (RIGHT), human responses are significantly shorter, averaging 10 fewer tokens per response compared to GPT-3.5's responses. Secondly, we note that LLM responses tend to use more common words, yielding better fluency and generally smoother text compared to human responses.

These characteristics present challenges in ranking responses from diverse sources. Human responses, due to their brevity and unique word choice, often have lower length-normalized log probabilities than model responses. This discrepancy leads to many margin violations during training using Eq. (2), and more parameter updates to ensure human responses score higher than model outputs.

To mitigate this, we assign a length scaling factor $\lambda$ of 0.85 to all model responses, including those from Llama-2-7B and GPT-3.5-turbo-0613, maintaining a $\lambda$ of 1.0 for human responses. This effectively shifts the log probability score distributions for model responses closer to human ones (Figure 7), reducing margin violations. We are also exploring adjusting the margin size and curriculum learning, which gradually increases the difficulty of training samples to reduce violations, as potential directions for future research.

## 6    CONCLUSION

We introduce RRESCUE, a novel training paradigm for optimizing LLMs with ranking metrics. This method finetunes LLMs to distinguish between sound and flawed rationales, and to recognize rationales that are grounded in the relevant context sections, thereby enhancing overall accuracy in various tasks. We experiment with partial ordering of candidate responses, showing it to be more robust and less susceptible to noise. RESCUE exhibits competitive performance on recent benchmarks, indicating promising avenues for future research in this area.

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
