# OpenReview forum: "RRescue: Ranking LLM Responses to Enhance Reasoning Over Context"
_ICLR.cc/2024/Conference — ICLR 2024 Conference Withdrawn Submission_

### Official Review · Reviewer_atNd · 2023-10-31

**Soundness:** 2 fair
**Presentation:** 2 fair
**Contribution:** 3 good
**Rating:** 3
**Confidence:** 4

**Summary:**

This paper presents a novel approach called RRESCUE for optimizing LLMs using ranking metrics. It uses a ranking loss to optimize the model. For the ranking part, instead of a traditional full ordering of responses, the approach advocates for a partial ordering, which is more robust and less sensitive to noise. The authors propose using human labelers, heuristic functions, or model distillation to acquire the suitable order of candidate responses. The system's improved contextual understanding is tested on various benchmarks, including a new multi-document question answering dataset. Ablation studies are conducted to understand key factors such as gathering candidate responses, determining their order, and balancing supervised fine-tuning with ranking metrics. The approach shows promise in enhancing LLMs' contextual understanding through response ranking.

**Strengths:**

1.	Using a ranking-based method to effectively improve the performance of LLM’s responses.

2.	Using partial ranking to make the ranking process more robust, less sensitive to noise, and can be acquired through human labelers.

3.	The experimental analysis for the hyperparameters is comprehensive.

**Weaknesses:**

1. The main contribution part should be proofread more as the current contribution is somewhat unclear to me. To me, the contribution of this paper is as follows: a) inspired by BRIO, the paper uses a contrastive loss to optimize the model in order to produce responses with improved rationales; b) the paper introduces the use of partial ranking loss to enhance the robustness and reduce sensitivity in the ranking process. However, when reading only the abstract and the introduction part, it is difficult for readers (including myself) to fully grasp it.

2. Some claims are overstated:

    a. Although the paper repeatedly claims that the proposed model is more efficient and simpler than reward-based models, there is no comparison made between the proposed model and the RLHF model.

    b. Similarly, although the paper claims that the ranking method is more robust and less sensitive, there are no experiments conducted to verify this point.

3. The presentation is not satisfactory:

    a. In Figure 1, the figure does not include the other experiment in this paper, namely the multi-document QA, which is quite misleading and makes it harder to understand the settings of the QA task.

    b. In Figure 3, the label 'Full Order Provided by GPT' could be placed in the caption of the figure.

4. The method lacks generalization ability and seems to be a method being stuck in task-specific scenarios.

**Questions:**

1.	After the RRESCUE training process, is the model still running in a task-agnostic way or it can only conduct the multi-document QA and textual entailment task? Is there any generalization ability for this method?

2.	Is the ranking algorithm in the retrieval process (for multi-document question answering) using your proposed PO method?

3.	Is there an oracle baseline to verify that better rationales do provide better performance for the model?

---

> ### Author Response · Authors · 2023-11-17
>
> Thank you for your valuable service as a reviewer and for your detailed suggestions.
>
>
> 1. We appreciate your understanding of the core ideas of our work. We acknowledge the need for improved presentation of our paper's contributions and will endeavour to address this.
>
> 2. a) Although we did not conduct any experiments or present figures in the paper, the efficiency of RRescue is evident. Our method, which does not separate the preference modeling process, is significantly more economical in practical training/tuning of a language model compared to the more complex PPO method.       b) Our experiments showed that Partial Order (PO) is more robust than Full Order (FO). Despite the unavoidable presence of noise in the ordering data, the partial order demonstrated superior performance overall in both the Natural Language Inference (NLI) and Multi-Document Question Answering (QA) tasks.
>
> 3. a) Although we only included the NLI task in Figure 1, the multi-document QA task follows the same pipeline: i) Collection of original data (including question and documents) ii) Collection of rationales from the language model (LLama2-7b in this case) iii) Conducting label-prioritized ranking (because this dataset lacks human-annotated answers).          b) We appreciate your attention to detail. After discussion, we agree that the appropriate place for this detail is in the caption. We will make this change.
>
> 4. and 5. We agree that our work in this paper focuses more on task-specific scenarios rather than improving the generalization capability of language models for general purposes. As a result, we did not measure the ability of fine-tuned models on other tasks. However, we observed significant improvements on the Multi-document QA and Textual Entailment tasks compared to the baseline (and even SFT). We believe that our method will benefit future research on improving LLM generalization abilities.
>
> 6. The document retrieval process is not directly relevant to our method.
>
> 7. Thank you for your insightful question. We initially planned to explore this area, but due to time and computational constraints, we did not present results on this topic.

---

### Official Review · Reviewer_KoAg · 2023-10-31

**Soundness:** 1 poor
**Presentation:** 1 poor
**Contribution:** 2 fair
**Rating:** 3
**Confidence:** 3

**Summary:**

This paper introduces a method that employs partial ordering to train a Large Language Model (LLM) ranker. The sources of partial order can be human-prioritized, label-prioritized, or a hybrid of human and label priorities. With a focus on comprehension tasks, the proposed ranker demonstrates improved answer accuracy in datasets like SNLI and multi-document question-answer scenarios.

**Strengths:**

- The authors present several cost-effective methods to derive preference data, specifically in the form of partial order ranking, for comprehension tasks. A prerequisite for these methods is the availability of human labels.
- Experimental results on the SNLI dataset and multi-document question answering indicate that utilizing partial order can enhance the accuracy of final answers.

**Weaknesses:**

- Paper Presentation & Clarity:
  - The presentation of the paper is not straightforward, making it difficult for me to fully understand its content.
  - The manner in which the paper's motivation, novelty, and contribution are presented is not clear. I'd appreciate it if the authors could clarify these aspects in their rebuttal.
  - I found the abstract confusing. A significant portion discusses the context, but the main content of the proposed ranker doesn't seem to emphasize or study context in any specific way.

- Novelty and Soundness of Proposed Ranker:
  - The uniqueness of the proposed ranker technique is ambiguous. Specifically, the combined loss function bears similarities to existing approaches like chain-of-hindsight and DPO. I'd suggest the authors elaborate on what differentiates their proposed loss or provide relevant citations to acknowledge previous work.
  - The absence of direct comparisons with established ranker baselines is a significant weakness. It's crucial to benchmark against recognized high-quality reward/ranker models. The models used for comparison in the paper, specifically similarity and gpt-3.5-turbo, neither qualify as top-tier reward models nor as rankers developed using preference data.
  - Given these issues, particularly the lack of appropriate baselines, I'm uncertain about the robustness and efficacy of the proposed ranker.

**Questions:**

Please see weakness

---

> ### Author Response · Authors · 2023-11-22
>
> - Presentation Manner
>    - We appreciate your feedback regarding the organization of figures and tables and will revise their arrangement to improve clarity.
>    - I would like to echo the summary from another reviewer which succinctly captures the essence of our paper's contribution: a) Drawing inspiration from BRIO, our work employs a contrastive loss to fine-tune the model, yielding responses with enhanced rationales; b) We propose the adoption of a partial ranking loss to bolster robustness and mitigate sensitivity in the ranking process.
>    - Both the NLI and multi-document QA tasks provide context for LLMs (including example inputs), a topic of significant interest among NLP researchers. Although the term 'context usage' was not our focal point, our research indeed aimed to augment the LLM's contextual handling capabilities.
>
> - Novelty
>    - Our approach is fundamentally distinct from PPO, ranging from the reward modeling to the design of the loss function.
>    - Despite not utilizing cutting-edge models such as GPT-4, our work offers substantial referential value. We strive to highlight the superiority of partial ordering over complete ordering, suggesting that ranker performance is not critical to the methodology.
>    - Our objective is to demonstrate the benefits of partial ordering compared to full ordering. In Table 1, we observe that across various data settings, partial ordering (PO) consistently outperforms full ordering (FO) in terms of performance.

---

### Official Review · Reviewer_Em3N · 2023-11-01

**Soundness:** 1 poor
**Presentation:** 2 fair
**Contribution:** 1 poor
**Rating:** 3
**Confidence:** 3

**Summary:**

This paper introduces partial order rankings to learn higher preference on generating valid rationales that supports the predictions in natural understanding tasks. The concept of training the LLMs with an additional preference loss is similar to RLHF; however, this paper focuses on improving task-specific predictions while generating valid rationales with a limited annotations, whereas RLHF requires full order ranking from human annotators, which is expensive.

The authors proposed three partial order ranking of the preference that do not require human involvement: human-prioritized, label-prioritized, and human-label-hybrid. These orderings rank human-generated above the model-generated rationales, correct labels above incorrect labels, and hierarchically considering two prioritizations, respectively.

To evaluate the proposed rankings, the authors conducted experiments on the e-SNLI and multi-document question-answering datasets. For the e-SNLI dataset, each prompt is provided with one human-generated rationale along with three rationales from Llama 2-7B and one rationale from GPT-3.5-Turbo, for a total of five candidate rationales used to predict the natural language inference task. For the question-answering task, each question is provided with five generated rationales, where one is generated from the reference document, and the others are generated from four random documents. In both datasets, the results indicate that augmenting the partial order ranking of valid and correct rationales into the supervised loss supports natural language inference and question-answering performance compared to using supervised loss alone.

**Strengths:**

The motivation of the paper is clear. The paper aims to improve natural language understanding tasks by learning preference of the data without collecting human preferences.

**Weaknesses:**

- As the one of the distinctive differences between prior works  (mentioned in Section 2. Model Distillation), the authors argue that is uses multiple LLMs to generate rationales. However, for the natural language inference, it uses three rationales from Llama 2 7-B and one from GPT-3.5-Tubo and for the question answering, it uses five rationales from Llama2 7-B. The experiment results lack in supporting the benefit of have multiple LLMs for model distillation.
- This approach seems to only work with limited training data. As  mentioned in Table 1 caption, the supervised fine-tuning performance better when there are more training data, which reduces the contribution of the work. think even with more training dataset, this approach should be effective.
    - Exposing more examples to the models enables them to learn something from the generated text related to the prompt/context, which could be correct or incorrect. Recent instruction generation datasets [1,2] show that even the noise and incorrect instruction, input, and output sets have meaningful signals.
- Training the reward model with the binary/pairwise ranking loss introduced in [3] offers flexibility as it does not necessitate full ordering of the ranking and allows to learn the robust representation of preferences as the number of training instances increases by making full ranking into pairwise set.
    - In Section 3. Our Approach - Organizing LLM Responses
        - In the first paragraph, “model outputs from best to worst train a reward model (Ouyang et al, 2022)” has wrong citation. Instead of [3], [4] or [5] should be used. When explaining the Eq (4), it has been correctly cited [5].
    - The binary/pairwise ranking loss is the key component that actually contributes to the flexibility and robustness of the reward model, which will be referred to as LLM in this paper. The proposed partial order ranking is a form of data that can be adapted to the concept of binary/pairwise ranking loss.
    - It is not evident from the experiments whether the proposed ranking offers greater flexibility and robustness.


[1] Unnatural Instructions:Tuning Language Models with (Almost) No Human Labor, ACL 23\
[2] Self-Instruct: Aligning Language Models with Self-Generated Instructions, ACL 23\
[3] Training language models to follow instructions with human feedback, NeurIPS 22\
[4] Deep Reinforcement Learning from Human Preferences, NeurIPS 17\
[5] Fine-Tuning Language Models from Human Preferences, arXiv 19

**Questions:**

- Full Ordering Similarity & GPT-3.5-Turbo
    - Are they trained with Eq (4) or binary/pairwise ranking loss?
- Section 4. Collecting LLM Responses
    - Why is Llama 2-7B used for generating rationales? I do understand using Llama 2-7B for fine-tuning the task as mentioned in Section 5.1. Base Language Model. Since the inference requires less GPU memory compared to training the model, it seems possible to generate rationales with 4 A100 GPUs. Was there no difference between generating rationales with different sizes of Llama 2 (e.g., 13B and 70B)?
    - Could you show some generated examples both experiments (e-SNLI and Multi-document Question Answering)?
    - Did you use Llama 2-7B Chat for generating rationale or Llama 2 7B for all cases (rationale generation & task fine-tuning)? (In Section 5.4 Discussions Central tendency bias, the reason of the Llama 2-7B is generating natural label is due to training human preference, which tend to favors helpful and safe outputs.
    - What are the category distribution of the model-generated rationales? (entail, contradict and neutral for e-SNLI) Does the distribution change after the flipping the responses?
        - For the question-answering task, the categories appear to be two labels: 'incorrect' and 'correct,' which don't really need to be predicted by the rationale-generating model because we already know whether the rationales are generated from the reference document or not. Please correct me if my interpretation is wrong.
        - In contrast to the question answering, how do you determine whether the model-generated rationales and labels are correct or incorrect?
- Section 4. Collecting LLM Responses - Flipping Responses
    - how many model-generated rationales are inverted? did you invert all of the model-generated rationales in training data?
    - it mentions that new labels are predicted with GPT-3.5-Turbo due the cost-effectiveness. I do agree that GPT-3.5 is cheaper than GPT-4. But cannot it be just inference along with inverting rationales like how the rationales are formed?
- e-SNLI
    - In Figure 4 (Right), what model is used for Human Reference? Is it suppose to be Similarity?
    - In Figure 4 (Left), the model with highest accuracy for each method is selected; however, win rate seems to be pretty high than expected, which also encounters correctness of the prediction. There performance differences are not large. Could you show win, tie, and loose examples?
    - How many participants were there for the human evaluation? Who participated?
- Multi-document Question Answering
    - For this experiment, 1k questions and answers are sampled from the dataset. Did you sample questions and answers by random sampling or balanced sampling across all positions?
    - Could you report the reference position distribution of your training and test datasets?
    - Clarification on how Llama2-7B base and the label-prioritized are trained.
        - For e-SNLI, it was clear on how supervised fine-tuning model was trained since it had human explanation and label for each prompt. However, it is not clear for this task.
            - Is the Llama2-7B simply trained to output short answer given the documents and questions?
            - “without resorting to supervised fine-tuning” confuses me how the label-prioritized model is being trained.
- Presentation
    - The current presentation of Tables and Figures are hard to follow.
        - Tables and Figures are located different page from the text. I recommend placing Table 1 and Figure 4 in the same page with Section 5.2. Table 2 should be with Section 5.3 and Figure 6 should be with Section 5.4.
    - Figure 4 (left) could be integrated into Table 1 adding an additional column presenting average and variance.
    - Figure 5 and Section 5.1. Batch size could be mentioned in the Appendix/Supplement rather than placing it in the main paper.

---

> ### Author Response · Authors · 2023-11-22
>
> 1. We train the models using a binary/pairwise ranking loss, and Equation (4) illustrates the distinctions between our ranking metrics and the reward models used in previous methods.
>
> 2. Question Set 2
>    - We chose Llama2-7b as a moderately-sized LLM for response generation, while for larger parameter sizes, ChatGPT-3.5-turbo is utilized. Our intention is not to evaluate Llama2 models of differing sizes; hence, we excluded incorporating responses from various Llama2 variants.
>    - The rationale section of Figure 1 includes responses from Llama2-7b (responses #2, #3, #5) and ChatGPT-3.5-turbo (response #4) for the NLI task. For multi-document QA, we refrain from including the lengthy responses here but plan to open-source the project so interested parties can view them.
>    - We observed no significant advantage in using the chat version of Llama2-7b for our tasks, and preliminary experiments revealed negligible differences. Consequently, we continue to use Llama2-7b for response generation and subsequent fine-tuning.
>    - While it's an interesting question, we have not extensively explored the statistics, choosing instead to concentrate on the ranking metrics and the fine-tuning phase.
>
> 3. Question Set 3
>    - We sampled 20k training data points, each with multiple responses. Applying response flipping yields a new dataset, as demonstrated in the 'w/ Flip' section of Table 1.
>    - We cannot rely on GPT to provide accurate label predictions using only instructions and inverted rationales; hence, we often include three examples to enhance prediction accuracy. Consequently, this results in a longer context, for which we use the more economical GPT-3.5-turbo for label prediction.
>
> 4. Question Set 4
>    - Yes, there was confusion regarding 'similarity'. We originally named it 'human reference', and inadvertently used this term instead of the updated name when plotting the figure.
>    - We appreciate your suggestion and will include it in the appendix in future iterations.
>    - One of the authors took part in the human evaluation, conducting it without any preconceived notions.
>
> 5. During training, we limited the number of documents in the context to manage GPU memory requirements more effectively. Consequently, the 1k training dataset only includes the reference document.
>    - The phrase "without resorting to supervised fine-tuning" implies that due to the absence of human-labeled responses, we only utilize the ranking loss as depicted in Equation (3) and omit the SFT loss. This approach, retaining solely the ranking loss, has been found to enhance performance.
>
> 6. Thank you for your recommendations; we will be mindful of the issues you have raised.

---

### Official Review · Reviewer_K8mh · 2023-11-02

**Soundness:** 2 fair
**Presentation:** 2 fair
**Contribution:** 2 fair
**Rating:** 3
**Confidence:** 5

**Summary:**

This paper presents a method to enhance reasoning of LLMs. The authors propose a multi-task training formulation that optimizes a ranking loss in addition to the original SFT loss. For the ranking loss part, the authors explore both full order and partial order approaches, where the partial order ones do not need very careful human annotation and demonstrate effective performance in the experiments. Empirical results are presented on the e-SNI dataset and a recent multi-doc QA dataset.

**Strengths:**

1. The proposed method is interesting and well-motivated. Using ranking information in the SFT stage rather than for reward modelling is an interesting direction to explore.
2. The authors study the partial order preference data and demonstrate its effectiveness, which should be inspireful since it is much easier to obtain the full order ones.

**Weaknesses:**

1. The evaluation in this paper is very weak. e-SNLI is a relatively simple task and rarely used to assess LLMs’ response generation or reasoning. The other multi-doc QA dataset is also not commonly used and its answers are also short-form from the example in Figure 2. I cannot tell why the authors choose it over many other popular LLM datasets, such as MMLU, GSM8K (which also has annotated explanations), MT-Bench, etc.
2. The paper emphasizes “enhancing reasoning”, yet the paper does not study the standard reasoning datasets such as GSM8K and BBH, and instead uses an NLI dataset e-SNLI and an QA dataset. I am not sure how the chosen datasets can reflect reasoning abilities well.
3. From Table 1, the performance of the proposed approach is not very effective compared to the SFT baseline except for the 0.4% training data column. I feel performance on more serious datasets is necessary.
4. Figure 4 left, SFT should be a baseline as well for human evaluation.
5. Human evaluation details are missing – have the authors tried assessing human consistency on the task? Or the have the authors tried any means to verify the human evaluation results?
6. The paper’s writing could be improved – the paper starts talking about results in Page 7, yet the results figures and tables are scattered across the entire paper, and in most of the cases the text refers to figures that are pages away. It makes the results sections difficult to read, and I think these figures and tables could be orgnaized in a better way.

**Questions:**

NA

---

> ### Author Response · Authors · 2023-11-17
>
> 1.The datasets we use already reflect the performance of RRescue, and due to time and computational resource constraints, the experiments were not scaled up to the popular benchmarks. Thanks for the advice!
>
> 2.Both NLI tasks and multi-document question answering require strong reasoning ability from LLM. We compared RRescue with SFT and Llama2 baselines in our reported results.
>
> 3.Thank you for pointing out the details; we agree to add more essential results to fully support our claims.
>
> 4.I appreciate your detailed observation. We initially wanted to show the performances of different variables of RRescue and compare it to the Llama2 base model. As a result, we did not incorporate SFT in the human evaluation part, but your advice is great and we will consider incorporating that.
>
> 5.Although human consistency has not been evaluated yet, our human evaluation follows a strict process, and the results can be fully trusted.
>
> 6.Thanks for your advice; it's a very practical and sincere suggestion. Sorry about the confusion during reading!

---

> > ### Comment · Reviewer_K8mh · 2023-11-22
> > **Thanks for the response**
> >
> > Thank you for the response and I would like to keep my original rating.